# Analysis of MicroRNA-Transcription Factors Co-Regulatory Network Linking Depression and Vitamin D Deficiency

**DOI:** 10.3390/ijms25021114

**Published:** 2024-01-17

**Authors:** Maria Sala-Cirtog, Ioan-Ovidiu Sirbu

**Affiliations:** 1Department of Biochemistry and Pharmacology, Discipline of Biochemistry, University of Medicine and Pharmacy “Victor Babes”, E. Murgu Square No. 2, 300041 Timisoara, Romania; ovidiu.sirbu@umft.ro; 2Center for Complex Network Science, University of Medicine and Pharmacy “Victor Babes”, E. Murgu Square No. 2, 300041 Timisoara, Romania

**Keywords:** depression, vitamin D deficiency, transcriptome, GEO2R, bioinformatics, microRNA, transcription factors

## Abstract

Depression and vitamin D deficiency are often co-occurring pathologies, the common pathogenetic ground of which includes an augmented inflammatory response. However, the molecular details of this relationship remain unclear. Here, we used a bioinformatic approach to analyze GEO transcriptome datasets of major depressive disorder (MDD) and vitamin D deficiency (VDD) to identify the hub genes within the regulatory networks of commonly differentially expressed genes (DEGs). The MDD-VDD shared regulatory network contains 100 DEGs (71 upregulated and 29 downregulated), with six hub genes (PECAM1, TLR2, PTGS2, LRRK2, HCK, and IL18) all significantly upregulated, of which PTGS2 (also known as COX2) shows the highest inference score and reference count. The subsequent analysis of the miRNA-transcription factors network identified COX2, miR-146a-5p, and miR-181c-5p as key co-regulatory actors in the MDD-VDD shared molecular pathogenic mechanisms. Subsequent analysis of published MDD and VDD transcriptome data confirmed the importance of the identified hub genes, further validating our bioinformatic analytical pipeline. Our study demonstrated that PTGS2 was highly upregulated in both depressive patients and patients with low vitamin D plasma levels. Therefore, regulators targeting PTGS2, like miR-146a-5p and miR181c-5p, may have great potential in controlling both diseases simultaneously, accentuating their role in future research.

## 1. Introduction

As one of the most frequent psychiatric diseases, depression affects 5% of adults worldwide and is ranked the fourth contributor to the global disease burden [1]. Depression increases the risk of suicide but also of chronic disorders such as cardiovascular diseases, diabetes, or cancer [2,3,4,5]. Conversely, individuals affected by chronic illnesses may develop a depressive state due to the difficulties associated with their condition.

Even though the etiology and pathophysiology of depression are still not fully understood [6], the monoamine hypothesis, along with the dysregulation of the hypothalamus–pituitary–adrenal (HPA) or hypothalamus–pituitary–gonadal (HPG) axes, remain the most accepted theories to be associated with the pathophysiology of depression [7]. Reduced levels of norepinephrine, dopamine, and serotonin in the brain of depressed patients can alter several functions, including the regulation of emotions, energy levels, and stress response [8,9]. In recent years, the role of other molecules like cytokines, neuropeptides, and oxidative stress biomarkers has been linked to depressive disorder [10,11].

Despite the impressive advances in therapy, nearly 60% of depressive individuals receive no treatment or discontinue their medication [12]. This phenomenon is due to the rather modest investments in mental health care, the social stigma associated with mental disorders, the financial burden, and the lack of education in the general population, manifested by the fear of side effects or addiction or the belief that medication is unnecessary [13,14]. We acknowledge that MDD is the most frequent comorbidity of schizophrenia (SZ); moreover, epigenomic mechanisms have been hypothesized to contribute to the development of MDD/SZ [15]. Nevertheless, despite the numerous studies in the field, the prevalence of MDD in SZ is not known [16]. There is a consensus on the need to broaden the therapeutic options for depression. Recently, attention has been increasingly drawn to dietary factors [17], and Omega-3 fatty acids, minerals like Zinc or Magnesium, vitamin D, and the vitamins of the B group have all been considered therapeutic adjuvants [18]. In general, appropriate nutritional micronutrients are essential for preventing the risk of depression [19]. Among them, the potential role of vitamin D in depression has received increasing research focus [20].

Accumulating evidence shows that low plasma levels of vitamin D are a common finding in patients with depression; however, it is still unclear whether vitamin D deficiency is a determining factor or has a simple association with depressive disorders [21]. An explanation could be that some symptoms of depression include withdrawal from social activities. Indoor confinement, a common state in depression, reduces direct exposure to sunlight, diminishing vitamin D synthesis in the skin [22]. While the link between depression and hypovitaminosis D continues to accumulate solid clinical evidence, the molecular pathogenic mechanisms linking the two remain elusive.

An indoor lifestyle has raised vitamin D deficiency (VDD) to a global public health status, impacting a billion people worldwide [23]. Mostly known for its role in bone health, the fat-soluble “sunshine” vitamin [24] can also act as a neuroactive steroid with anti-inflammatory and antioxidant properties. A large-cohort 2019 study on vitamin D supplementation reported a significant improvement in multiple inflammation-related biomarkers [25]. Interestingly, neuroinflammation was documented in one in four patients with MDD [26]. In a large cohort study, low levels of 25(OH)D (<14 ng/mL) were associated with the existence and severity of depressive disorder, implying that VDD may represent a biological vulnerability to depression [27].

Here, we analyzed public transcriptome datasets from individuals diagnosed with MDD and VDD, aiming to identify the hub genes within the regulatory networks of commonly differentially expressed genes (DEGs). GEO (Gene Expression Omnibus) microarray datasets [28] were downloaded and subjected to GEO2R analysis, followed by identification of the common differentially expressed genes. The shared DEGs network was next subjected to protein–protein interaction (PPI) and pathway analysis to reveal the connected mechanism between the two conditions. Finally, we constructed a network of transcriptional (transcription factors—TFs) and post-transcriptional (microRNAs) co-regulators of the identified hub genes.

To the best of our knowledge, this is the first attempt to explore the relationship between depressive disorder and VDD at the transcriptional level, focusing on co-regulators of the common hub genes. The hub-genes co-regulatory network might serve as novel putative targets relevant to the clinical and therapeutical management of depressive disorder.

## 2. Results

### 2.1. Identification of DEGs

The logical succession of the analytical steps of our study is represented in the flowchart depicted in Figure 1. Basic information concerning the eight depression GEO datasets (GSE190518, GSE98793, GSE76826, GSE217811, GSE101521, GSE23848, GSE80655, and GSE169459) and two hypovitaminosis D datasets (GSE157939 and GSE22523) are shown in Table 1.

GEO2R analysis of depression transcriptome data retrieved 7353 upregulated and 6564 downregulated genes. A similar analysis of hypovitaminosis D data identified 210 upregulated and 119 downregulated DEGs, of which 100 are shared between the two pathologies: 71 upregulated and 29 downregulated (Figure 2 and Appendix A). 

### 2.2. PPI Network and Functional Enrichment Analysis of Common DEGs 

The depD list of 100 DEGs was uploaded on STRING, and the PPI network was constructed at a medium (0.400) confidence level; the network’s edges represent interaction evidence retrieved from all seven possible sources (text mining, neighborhood, experiments, gene fusion, databases, co-occurrence, and co-expression). The PPI network included 98 nodes and 89 edges with a statistically significant PPI enrichment (*p*-value = 0.01). The results were then imported to Cytoscape software version 3.9.1 for visual analysis (Figure 3).

GO enrichment analysis showed that this network is significantly enriched in biological processes related to the cellular response to stress, with a significant immunological component, like positive regulation of the defense and inflammatory response (GO:0050729), immune response (GO:0006955), and regulation in response to external factors and stress (GO:0080134) (Figure 4a). This suggests that the immune and inflammatory responses link the pathophysiology of the two conditions. A similar analysis of the KEGG pathways enrichment led to six distinct pathways, the most significant being metabolic pathways (hsa01100), followed by Alzheimer’s (hsa05010) and Parkinson’s disease (hsa05012) (Figure 4b). This suggests that the transcriptome changes shared by the two conditions might represent the molecular substrate of metabolic alterations related to neuronal degenerative processes. Details regarding GO enrichment analysis and KEGG pathways are provided in Appendix A.

### 2.3. Hub Genes Identification

In order to identify the hub genes within the PPI network, data were uploaded and analyzed using cytoHubba, a Cytoscape plug-in designed to identify a network’s critical genes according to their number of associations. The top 10 genes were selected in each of the seven analytical algorithms used, and the final six hub genes were identified in the uPSet diagram: PECAM1, TLR2, PTGS2, LRRK2, HCK, and IL18, all upregulated (Figure 5 and Table 2).

### 2.4. Construction of TF-Hub Genes-MicroRNA Regulatory Networks

Next, we used NetworkAnalyst to build the network (trimmed down to a minimum of two-degree connectivity) of transcriptional and post-transcriptional regulators of hub genes, focusing on TFs and microRNAs, respectively. The final network (20 nodes and 66 edges) included six miRNAs and nine TFs targeting five of the six hub genes (PECAM1, TLR2, PTGS2/COX2, LRRK2, and HCK) (Figure 6). Of note, PTGS2, the only hub gene controlled by both microRNA and TFs, has the highest (33) connectivity degree and, thus, represents a potential link in the clinical development of depression and hypovitaminosis D. Of the nine TFs, TFAP2A is predicted to concomitantly regulate three hub genes (PTGS2, PECAM1, and HCK), while the rest (ETS1, NFKB1, TCF3, STAT3, RELA, USF1, NFKB2, and CTCF) might each regulate two of the hub genes. Of the six microRNAs, hsa-mir-335-5p and hsa-mir-146a-5p display three-degree connectivity in the hub DEGs network, while hsa-miR-181b, hsa-miR-181c, hsa-miR-181d, and has-miR-543 simultaneously regulate two hub genes, PTGS2 and LRRK2 (Appendix A).

### 2.5. Data Validation

The expression of the six hub genes common in MDD and VDD was further validated in an independent human brain RNA-seq dataset, GSE125664. Except for HCK, all hub genes were significantly upregulated in depressive patients compared to controls (Figure 7).

Next, we interrogated the Comparative Toxicogenomic Database (CTD) for the interference score and the reference count of the six depD hub genes. We found that they are significantly connected to depressive disorder, major depressive disorder, osteoporosis, bone diseases, and vitamin D deficiency. This suggests that all six hub genes participate in multiple pathophysiological processes associated with both pathological conditions. Of note, the average hub gene inference scores are higher in depression compared to vitamin D deficiency, with the maximum for PTGS2 (Figure 8 and Appendix A).

As for transcriptional regulators validation, we observed that the expression of five of the identified TFs (ETS1, TFAP2A, NFKB2, CTCF, and RELA) is significantly altered in the blood samples of depression patients (datasets GSE217811 and GSE23848) (Table 3); of note, except for ETS1, all transcription factors show increased expression in patients vs. controls.

To further validate the identified microRNAs, we interrogated PubMed for microRNAs dysregulated in depressive patients (38 research papers) and hypovitaminosis D (19 research papers) (Appendix A). After removing the duplicates, we identified six commonly upregulated and ten commonly downregulated microRNAs (Figure 9).

Only two of these mutually deregulated microRNAs (miR-146a-5p and miR-181c-5p) are part of the TF-hub genes-miR network, where they are predicted to impact the expression of PTGS2. This suggests that the PTGS2 expression is the result of a delicate regulatory balance between TFs and microRNAs. The two microRNAs are downregulated in both depressed and vitamin D-deficient patients, which can explain the overexpression of the commonly targeted hub gene—PTGS2.

## 3. Discussion

The association between depression and vitamin D deficiency has long been an attractive topic for researchers worldwide. Clinical studies have shown that these two widespread pathological conditions go hand in hand [23], yet little is known about their common molecular mechanisms.

Herein, we interrogated public transcriptome data to identify and functionally characterize the gene expression networks commonly underlying depressive and vitamin D deficiency disorders in humans. To the best of our knowledge, this is the first attempt to investigate the MDD and VDD comorbidity mechanisms at the gene expression network level.

Our bioinformatic analysis was performed on eight depression and two vitamin D deficiency independent GEO gene expression datasets. We pooled the unique DEGs identified through GEO2R analysis of the MDD and VDD datasets and obtained 100 commonly deregulated DEGs: 71 upregulated and 29 downregulated.

Our gene ontology (GO) enrichment analysis of depD DEGs indicated four immune-related mechanisms putatively involved in both depression and hypovitaminosis D: positive regulation of the inflammatory response, immune response, regulation of external factors, and stress. This is in accordance with published data showing that depression and hypovitaminosis D pathogenic mechanisms involve inflammation and immune response [35]. Immune cells, like macrophages that produce pro-inflammatory cytokines such as IL12 and IL18, are involved in both MDD and VDD; it is thus conceivable that alterations in the inflammatory status in hypovitaminosis D precede/trigger the fluctuations in the psychosomatic dimensions of MDD [36].

Our KEGG enrichment analysis pointed towards neurodegenerative disorder pathways such as Alzheimer’s disease (AD) and Parkinson’s disease (PD). Prior studies indicated that depression could be linked to neurodegenerative processes, including dysfunctions of the dopaminergic, serotoninergic, and noradrenergic systems. Moreover, due to an overlap with other Parkinson-related symptoms, depression is often underdiagnosed and undertreated in patients suffering from both conditions [37]. In addition, neuroinflammatory processes play an essential role in both depression and dementia, indicating common pathways connecting Alzheimer’s disease with depression [38].

Interestingly, we have noticed an enrichment in the retrograde endocannabinoid signaling pathway, one of the main pathways accountable for stress, a well-known major risk factor for depression. The endocannabinoid system has been linked both physiopathologically and genetically to depressive disorders [39,40]; moreover, commonly used antidepressants downregulate the expression of cannabinoid receptors in the brain [41]. The endocannabinoid system is not only implicated in neuroinflammation but also chronic pain processing in the brain and gut. This is in accordance with a study of a rat model showing that low vitamin D levels amplify the inflammatory response by mediating endocannabinoid receptor signaling [42].

Our CytoHubba analysis identified six hub genes (PECAM1, TLR2, PTGS2, LRRK2, HCK, and IL18), all significantly upregulated in both depression and hypovitaminosis D, with PTGS2 having the highest inference score and reference count. Except for HCK, the expression of all hub genes was validated in a GEO transcriptome dataset from neurons isolated from MDD patients refractory to SSRI (GSE125664). The analysis of this external dataset confirmed that PTGS2 was the most overexpressed hub gene in depressive patients versus control.

Prostaglandin-endoperoxide synthase 2 (PTGS2), also known as Cyclooxygenase 2 (COX-2), is involved not only in inflammation but also in the central nervous system neurodegenerative processes [43]. The relationship between depression and inflammation is not a new topic. However, regardless of the type of study, COX2 seems to be the key element of the dysregulated inflammatory response in depressive patients [44]. This provides a solid ethio-pathogenic argument for the use of anti-inflammatory agents for controlling depressive symptoms in MDD. Moreover, a meta-analysis stipulated that the symptoms of depression could be controlled with Celecoxib, a non-steroidal anti-inflammatory drug (NSAID) presenting high COX-2 selectivity, without developing any serious side effects [45].

Based on our results, PTGS2 (COX2) was highly upregulated in both depressive patients and patients with low vitamin D plasma levels, as shown in the present study for the first time. In this regard, vitamin D significantly decreased COX-2 expression in brain areas and inhibited the degranulation of activated neutrophils via ROS reduction [46]. In this respect, adding COX-2 inhibitors and vitamin D supplementation can significantly improve the effect of regular antidepressive regimens.

LRRK2, a major modulator of neuroinflammation involved in the pathogenesis of Parkinson’s disease, is upregulated in both depressive (logFC = 0.56) and VDD (logFC = 2.57) patients [47,48]. LRRK2 mutations have been found in individuals without manifest Parkinson’s disease, in which compensatory changes in the serotonergic system were described [49]. Direct inhibition of LRRK2 by PF-06447475 significantly reduces depression-related symptoms in mice, suggesting that LRRK2 might be a valuable therapeutic target in MDD [50]. In contrast, very little is known about LRRK2 and vitamin D deficiency, which is why we were intrigued that in both VDD datasets used in our study, LRRK2 is strongly upregulated. A meta-analysis estimated that patients with Parkinson’s disease had lower vitamin D levels than healthy controls, and it seems that this is reflected in the loss of dopaminergic neurons; however, whether this associates the downregulation of LRRK2 with a significant impact on PD pathogenesis remains to be evaluated [51].

To better understand the role of the six hub genes in the two pathologies, we focused on the factors that modulate their expression at transcriptional and post-transcriptional levels. TFs and microRNAs can regulate gene expression: TFs bind to DNA regulatory regions and modulate RNA transcription, while miRNAs bind to 3’UTRs of mRNA targets and influence their translation and stability. Moreover, these two regulators can act concomitantly to regulate the same gene expression or alter each other’s expression, making it difficult to evaluate their impact on a gene network expression [52]. Our NetworkAnalyst study predicted nine TFs that can concomitantly regulate at least two depD hub genes. Five of these TFs, namely ETS1, TFAP2A, NFKB2, CTCF, and RELA, are significantly deregulated in the blood samples of depression patients compared to controls (GSE217811 and GSE23848).

Of all transcription factors, ETS1 was the only one downregulated in our study, and one of the reasons could be that its overexpression counteracts the depressive symptoms through ERK1/2—GalR2 signaling, having an antidepressant-like effect in rodents [53].

We have identified TFAP2A (transcription factor AP-2 alpha) as a modulator of PTGS2, the neuroinflammation hub gene related to oxidative stress [54]. Of note, differential methylation of TFAP2A has been described in a twin study of early-onset major depression disorder [55]. Furthermore, TFAP2A is among the TFs associated with mood disorders in general and MDD in particular in humans [56].

NF-κB, which was significantly upregulated in our depressive samples, regulated two hub genes, PTGS2 and TLR2. NF-κB signaling is a critical player in depression and stress-related depressive-like behaviors, presumably through a positive feedback loop involving BDNF and the BDNF/NFKB feedback loop in depression [57,58]. Recent research identified that various types of cancers like breast cancer or prostate cancer, in which depressive symptoms are highly prevalent, are associated with genetic variations like NF-κB polymorphisms [59].

Inflammatory cytokines like IL12 and IL18 can activate NF-κB subunits and are critical for inducing an inflammatory response [60]. Vitamin D can repress the activation of NF-κB through VDR- IKKβ interaction and subsequent disruption of the TNFα-induced IKK complex [61].

Of note, PTGS2 strongly synergizes with NFKB2 and RELA; the two NF-kB transcription factors affect pain behavior by upregulating the expression of PTGS2 (COX-2) [62], providing a molecular basis for vitamin D modulation of COX-2 gene expression.

Our transcriptomic analysis showed six miRNAs potentially interacting with at least two of the hub genes, of which two have been experimentally identified as downregulated in MDD and VDD: miR-146a-5p (targeting PTGS2) and miR-181c-5p (targeting both PTGS2 and LRRK2).

miR-146a-5p is a major regulator of COX2 expression in multiple experimental setups through multiple mechanisms, including targeting PTGS2 [63,64]. Furthermore, exosome-packed, microglia-derived miR-146a modulates hippocampal neurogenesis, and the downregulation of miR-146a-5p improves adult neurogenesis impairments and alleviates rodent depression-related behaviors [65,66]. miR-146a expression is severely downregulated in MDD, correlates with the severity of the disease, and is upregulated as the patients respond to therapy [67,68].

miR-181c-5p is involved in inflammation by regulating the expression of pro-inflammatory cytokines like TNF-α, IL-6, IL-1β, IL-8, and NF-κB [69,70]. While the overexpression of miR-181c-5p in mice hippocampus reduced the expression levels of inflammatory cytokines, lower levels of miR-181c-5p in the serum may indicate cerebral vulnerability in elderly patients [71].

In conclusion, our bioinformatic network-based approach provides new biological targets along with ideas for the early diagnosis of major depressive disorder combined with vitamin D deficiency by revealing the physiopathologic mechanisms common to both pathologies. To the best of our knowledge, this is the first study to explore the relationship between MDD and VDD using transcriptome analysis. By constructing the miRNAs–TFs-hub genes network, we found six miRNAs potentially interacting with at least two of the hub genes, of which two have been experimentally identified as downregulated in MDD and VDD: miR-146a-5p (targeting PTGS2) and miR-181c-5p (targeting both PTGS2 and LRRK2). These findings are consistent with our study because the downregulation of microRNAs could explain the mechanism of action by which inflammation genes like COX2, but also PECAM1, IL18, or TLR2, are upregulated in both patients with depression and low vitamin D levels. Furthermore, it becomes apparent that the existence of one disease could increase the risk of developing the other.

## 4. Conclusions

Our miRNA-TFs-hub genes network advances several novel molecular targets like PTGS2, LRRK2, miR-146a-5p, and miR-181c-5p, which might contribute to the development of novel diagnostic, monitoring, and therapeutic strategies in (hypovitaminosis D-related) depression. We propose these potential biomarkers to the medical research community as potential targets. The function of these hub genes and their post-transcriptional regulators needs to be further analyzed in an in vitro and in vivo model, which we will focus on in the near future.

## 5. Materials and Methods

### 5.1. Data Collection and Identification of DEGs

We interrogated the GEO database [72] using the keywords “depression” and “vitamin D deficiency”; data obtained from non-human specimens were excluded. Eight depression transcriptome datasets (GSE190518, GSE98793, GSE76826, GSE217811, GSE101521, GSE23848, GSE80655, and GSE169459), as well as two vitamin D deficiency datasets (GSE157939 and GSE22523), were included in the GEO2R analysis www.ncbi.nlm.nih.gov/geo/ge2r (accessed on 11 July 2023). DEGs with logFC (fold change) ≥|0.5| and a Bonferroni-adjusted *p*-value < 0.05 were considered statistically significant.

After excluding duplicates and genes with conflicting expressions, the depression and hypovitaminosis D (depD) datasets were compared, and the shared DEGs were selected for further analysis.

### 5.2. Construction of Protein–Protein Interaction Network and Functional Enrichment Pathway Analysis 

We used STRING 12.0 [73] to construct the protein–protein interaction (PPI) network from the depD dataset at a medium (0.400) confidence interaction score threshold. Gene ontology (GO) enrichment and Kyoto Encyclopedia of Genes and Genomes (KEGG) enrichment pathways analyses of the STRING PPI network were performed (FDR < 0.05) to identify molecular functions and pathways shared between depression and hypovitaminosis D. The depD PPI network was further imported and visualized in Cytoscape, version 3.9.1 [74].

### 5.3. Hub Genes Selection 

We used the Cytoscape plug-in CytoHubba to identify the significant hub genes in the depD PPI network, applying seven node-ranking methods (MCC—maximal clique centrality, degree, closeness, betweenness, bottleneck, radiality, and stress). We used an UpSet plot generated with the UpSetR package (version 1.4.0) [75] rather than a Venn plot to better display the intersections between the seven datasets. The top 10 hub genes, ranked according to all seven algorithms, were then visualized, and the common hub genes were identified.

### 5.4. Construction of Transcription Factors (TFs) and MicroRNAs (miRNAs) Regulatory Networks 

Next, we were interested in identifying the transcriptional (TF) and post-transcriptional (microRNA) regulators of depD hub gene expression. NetworkAnalyst version 3.0 [76] was first used to construct TF-gene regulatory network (based on the ENCODE database https://www.encodeproject.org/; accessed on 30 January 2019), followed by RegNetwork database (http://www.regnetworkweb.org; accessed on 2 April 2019) interrogation to generate TF-hub genes-miRNA co-regulatory interactions. In order to create a strongly interconnected network, only TFs and miRNAs common to a minimum of two hub genes (2-degree connectivity) were taken into consideration.

### 5.5. Data Validation 

We validated the depD hub genes by interrogating an independent human brain RNA-seq dataset from neurons derived from SSRI (serotonin reuptake inhibitor)-resistant MDD patients (GSE125664) and the Comparative Toxicogenomic Database (CTD), where the interference score and reference count between the hub genes and five pathologies related to depression and hypovitaminosis D were calculated.

To validate the microRNAs identified in the TF-hub genes-miRNA network, we interrogated Pubmed using the terms “depression”, “microRNA”, “low vitamin D”, “hypovitaminosis D”, “transcriptome regulation”, and the following inclusion criteria:Case–control, human studies on patients diagnosed with any form of depression or low (non-drug-related) vitamin D;Research articles reporting correlations between depression/vitamin D deficiency and differentially expressed microRNAs (with an adjusted *p*-value < 0.05);Full-text articles in English.

We excluded: Studies evaluating the effect of certain antidepressants or vitamin D supplementation on microRNA expression;Cancer-related research articles;Articles with insufficient data for subsequent analysis.

The list of microRNAs commonly deregulated in depression and in hypovitaminosis D was retrieved and, after removing the duplicates, compared to those identified in our depD TF-hub genes-microRNA regulatory network.

## Figures and Tables

**Figure 1 ijms-25-01114-f001:**
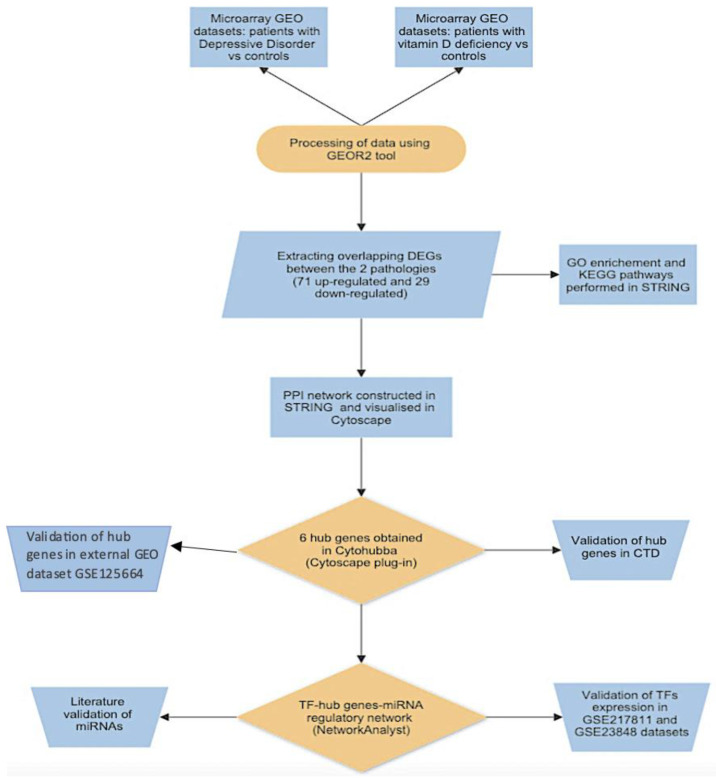
Research design flowchart diagram that represents the logical succession of the analytical steps of our study.

**Figure 2 ijms-25-01114-f002:**
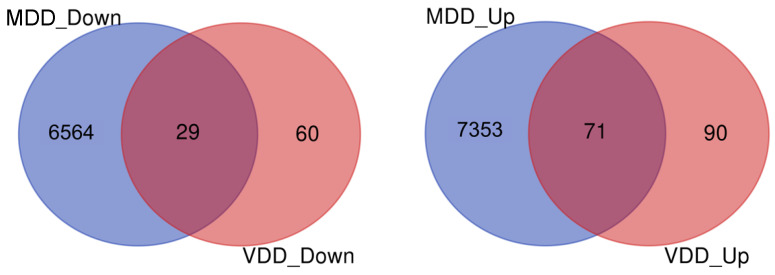
Venn diagram of downregulated (**left**) and upregulated (**right**) differentially expressed genes in major depressive disorder—MDD (blue) and vitamin D deficiency—VDD (red).

**Figure 3 ijms-25-01114-f003:**
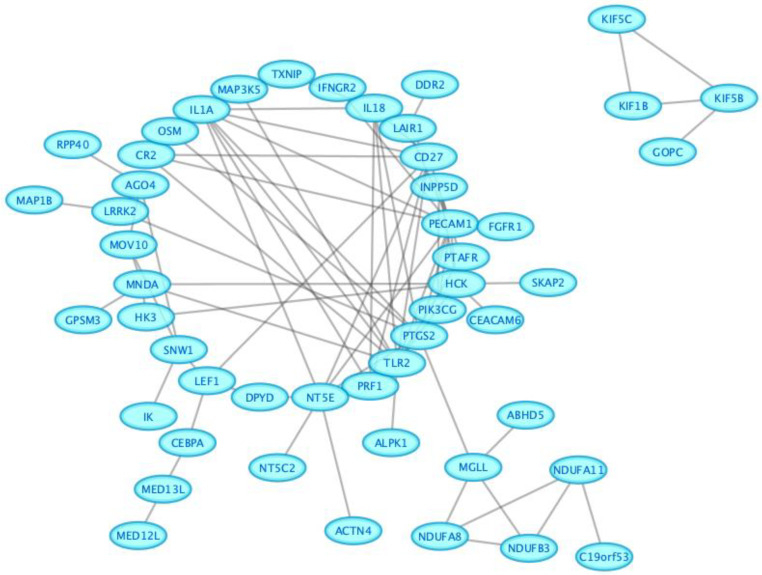
PPI network of the common DEGs between depression and vitamin D deficiency in Cytoscape; the network included 98 nodes and 89 edges with a statistically significant PPI enrichment (*p*-value = 0.01).

**Figure 4 ijms-25-01114-f004:**
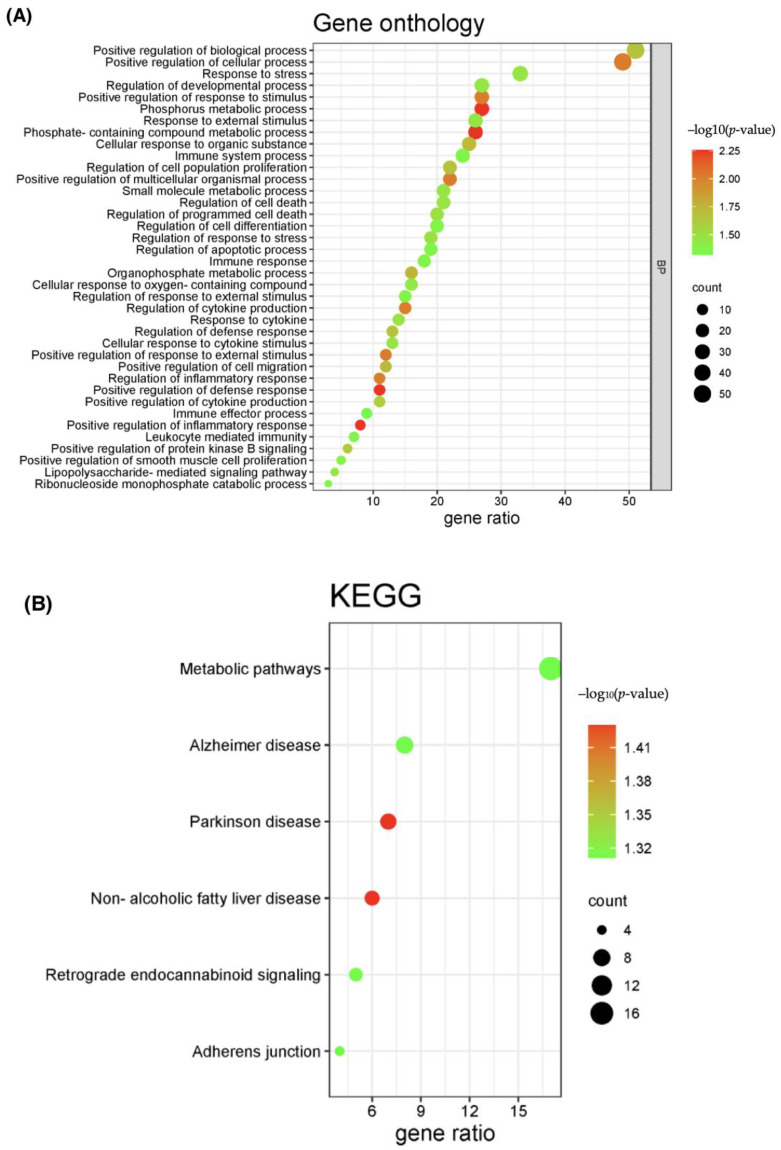
GO and KEGG enrichment analysis of common DEGs. (**A**) The enrichment analysis results of gene ontology biological processes. (**B**) The enrichment analysis results of KEGG pathway. Only results with adjusted *p*–value < 0.05 were considered significant.

**Figure 5 ijms-25-01114-f005:**
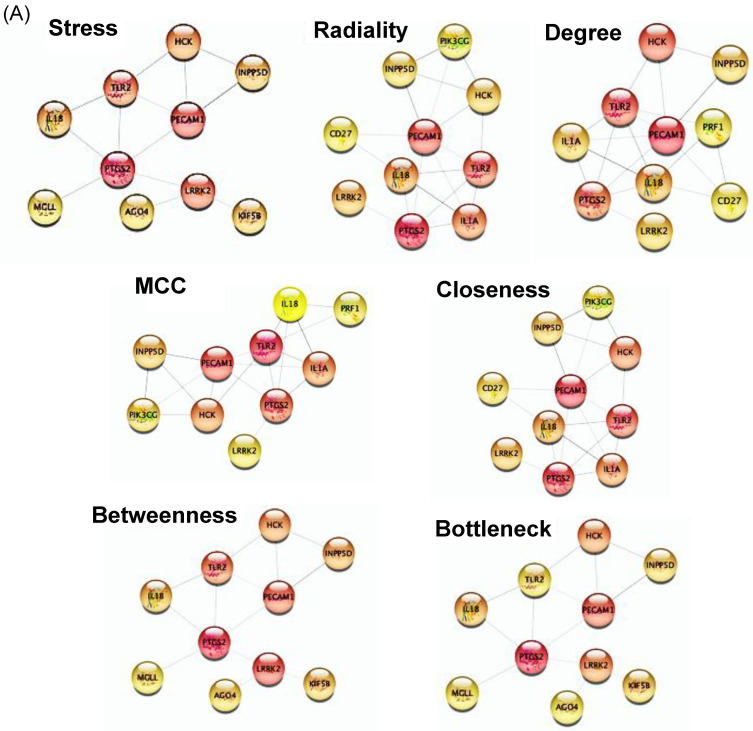
Hub genes identified in Cytohubba by using different algorithms and UpSet diagram. (**A**) Top 10 hub genes identified using seven different algorithms. (**B**) The vertical UpSet diagram shows six overlapping hub genes. The matrix columns correspond to the hub gene sets; the rows correspond to the intersections.

**Figure 6 ijms-25-01114-f006:**
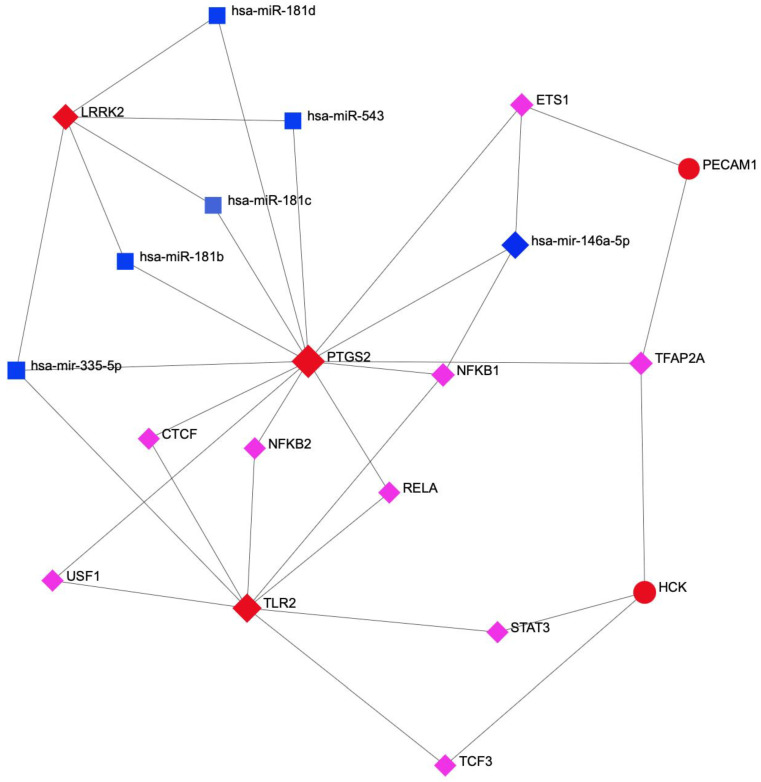
Integrated TF-miRNA-Hub interaction network. Red circles represent the five interconnected hub genes; blue squares represent the miRNAs, and pink rhombs represent the TFs.

**Figure 7 ijms-25-01114-f007:**
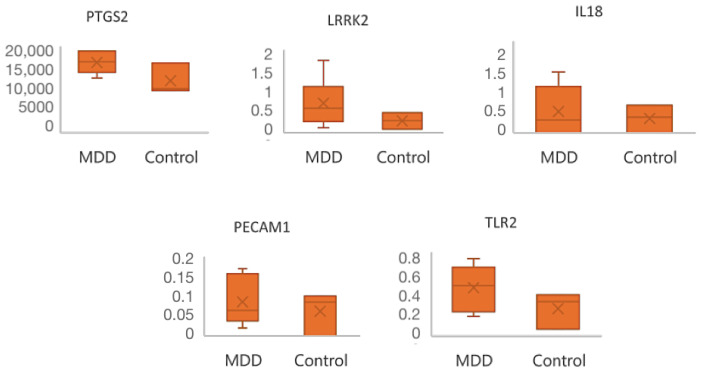
The validation of hub genes in external dataset GSE125664, in which the expression level of each hub gene from the MDD group was compared against the control group (healthy patients).

**Figure 8 ijms-25-01114-f008:**
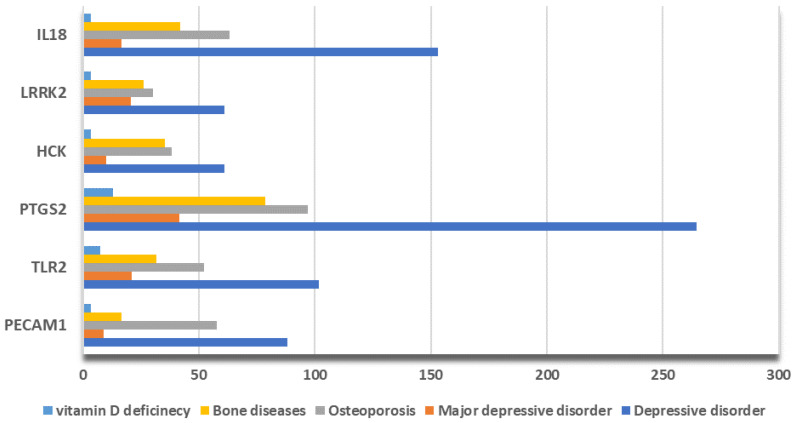
The correlations between hub genes, depression-related diseases, and low vitamin D-related diseases and their interference score in Comparative Toxicogenomic Database, with PTGS2 having the highest interference score in all mentioned pathologies.

**Figure 9 ijms-25-01114-f009:**
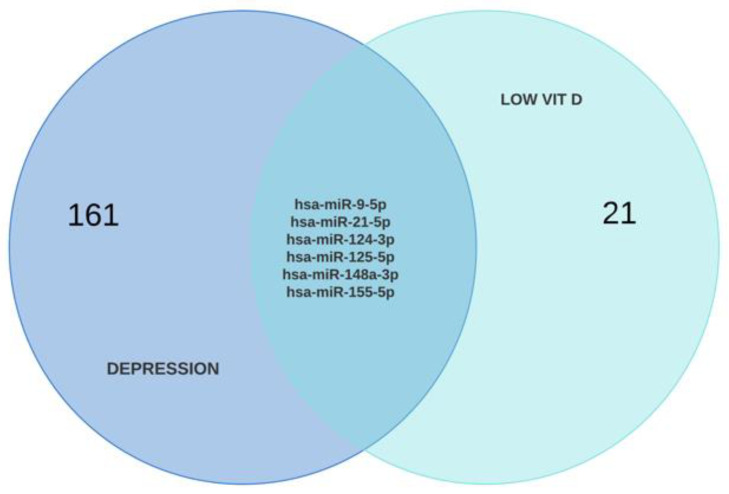
Venn diagrams of the miRNAs identified upon PubMed search: upregulated (blue) or downregulated (purple) miRNAs common to both depression and low vitamin D levels.

**Table 1 ijms-25-01114-t001:** GEO datasets’ basic information: number of participants, type of biological sample, pathological condition, and analytical platform used.

GSE Number	Participants (Patients/Controls)	Biological Sample	Pathological Condition	Analytical Platform
GSE190518	3/3	Blood	Depression	Illumina HiSeq 4000
GSE98793	128/64	Blood	Depression	Affymetrix GeneChip system
GSE76826	10/12	Blood	Depression	Agilent microarray
GSE217811	10/10	Plasma	Depression	Agilent microarray
GSE101521	18/38	Brain	Depression	Illumina MiSeq 2500
GSE23848	20/15	Blood	Depression	Sentrix Human-6 v2 Expression BeadChip
GSE80655	69/70	Brain	Depression	Illumina HiSeq 2000
GSE169459	3/3	Blood	Depression	Agilent microarray
GSE157939	80/80	Blood	Hypovitaminosis D	Fluidigm BioMark
GSE22523	2/2	Blood	Hypovitaminosis D	Affymetrix GeneChip system

**Table 2 ijms-25-01114-t002:** Hub genes’ functions.

Gene Symbol	Full Name	Function
PECAM1	Platelet And Endothelial Cell Adhesion Molecule 1	Cell adhesion molecule, which is required for leukocyte transendothelial migration (TEM) under most inflammatory conditions [29].
TLR2	Toll Like Receptor 2	Modulates the host’s inflammatory response and has been implicated in the pathogenesis of several autoimmune diseases [30].
PTGS2	Prostaglandin-Endoperoxide Synthase 2	With a particular role in the inflammatory response, during neuroinflammation, plays a role in neuronal secretion of specialized pre-resolving mediator [31].
LRRK2	Leucine Rich Repeat Kinase 2	Regulates neuronal process morphology in the intact central nervous system [32].
HCK	HCK Proto-Oncogene, Src Family Tyrosine Kinase	Plays an important role in the regulation of innate immune responses, including neutrophiles, monocytes, and macrophages [33].
IL18	Interleukin 18	Pro-inflammatory cytokine primarily involved in epithelial barrier repair [34].

**Table 3 ijms-25-01114-t003:** Validation of TFs in GSE217811 and GSE23848 transcriptome datasets.

adj. *p*-Value	logFC	TF	GEO Dataset
2.31 × 10^−2^	−1.18	ETS1	GSE217811
2.97 × 10^−2^	0.62	TFAP2A	
4.69 × 10^−2^	0.75	NFKB2	
2.91 × 10^−2^	0.55	CTCF	
8.78 × 10^−3^	0.58	RELA	GSE23848

## Data Availability

We would like to thank the GEO database for providing free open-source available data. All datasets analyzed in this study are available in the GEO repository at https://www.ncbi.nlm.nih.gov/geo/ (accessed on 11 July 2023).

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
