# Peer review of "Analysis of MicroRNA-Transcription Factors Co-Regulatory Network Linking Depression and Vitamin D Deficiency"

_ijms, 2024, doi:10.3390/ijms25021114_

Round 1
Reviewer 1 Report
Comments and Suggestions for Authors
The manuscript is interesting. The authors wanted to define the role of vitamin D in depression by analyzing available public microarray data. Overall, the manuscript is easy to read. However, improvements could be needed.
Minor point
- The quality of the figures needs improvement.
- English-language typos need to be corrected.
Major
- The manuscript also might need further width. How can you prove that these genes are contributing to the role of vitamin D in depression? Maybe you can perform further bioinformatic analysis with other tools to support your hypothesis.
- Other validation steps need to be taken, such as analyzing statistical depression datasets and checking if these genes come up as highly differential.
- The manuscript might benefit from more depth beyond the six hub genes. The main question posed by the authors, which is how vitamin D affects depression, is not yet clearly answered. Do these hub genes work together to increase or decrease the vitamin D effect? What is/are the pathway(s)? Did the results generate a hypothesis that could be tested experimentally in other papers? For example, how does vitamin D perform its functions? What are the receptors? which cells? How do these genes that you found interact with the vitamin D pathway?
- What is the translational aspect of your findings?
- English-language typos need to be corrected.
Reviewer 2 Report
Comments and Suggestions for Authors
Thank you for letting me review this article. The article analyzes the connections between vitamin D deficiency and Major Depressive Disorder.
Abstract line 21 further, please rephrase for clarity “from to conclude, …”
I suggest reworking the second paragraph of the introduction as it is too abstract and there have been new articles linking MDD to neuropeptides, cytokines and other molecules. Also, the HPA/HPG axes are long known to affect stress response and contribute to depression.
The whole introduction needs reworking for fact-checking as 75% of individuals receive no treatment or discontinue medication, which is untrue. The narrative suggests to the reader that vitamin D deficiency plays a pivotal role in depression or schizophrenia…
Look at the latest data https://evidence.nihr.ac.uk/alert/almost-half-people-long-term-antidepressants-stop-without-relapse/
Line 56. Depression is not a neuro-inflammatory disorder. Research suggests inflammation has a role in the etiology of depression.
concerning vitamin D deficiency – may it result from self-isolation in MDD?
Also, the order of sections needs revising, as materials and methods go after the introduction.
Comments on the Quality of English Languagetypos, cut sentences
Round 2
Reviewer 1 Report
Comments and Suggestions for Authors
No further comments
Comments on the Quality of English LanguageMinor corrections needed.
Author Response
Thank you very much for your positive feedback and for taking the time to review our manuscript.
Response to Comments on the Quality of English Language
Comment: Minor corrections needed Response: We have thoroughly revised the English used throughout the manuscript and corrected all typosReviewer 2 Report
Comments and Suggestions for Authors
The article is now much more suitable for publication.
I have one more suggestion, not mandatory - to include a few more references from the publishers' journals for lines 34-41 if such are available.
Author Response
Thank you very much for taking the time to review our manuscript.
Comment:
The article is now much more suitable for publication.
I have one more suggestion, not mandatory - to include a few more references from the publishers' journals for lines 34-41 if such are available.
Response:
Thank you for your helpful suggestion, we have indeed found interesting articles, that we have now mentioned - we kindly refer the reviewer to the revised version of our manuscript – lines 34-41 (reference 9 and 11)
[9] - Gold, P.W. The PPARg System in Major Depression: Pathophysiologic and Therapeutic Implications. Int. J. Mol. Sci. 2021, 22, 9248. https://doi.org/10.3390/ijms22179248
[11] Harsanyi, S.; Kupcova, I.; Danisovic, L.; Klein, M. Selected Biomarkers of Depression: What Are the Effects of Cytokines and Inflammation? Int. J. Mol. Sci. 2023, 24, 578. https://doi.org/10.3390/ijms24010578
Moreover, we found an interesting article regarding epigenomic mechanisms involved in the development of depression (reference 15 from the revised version of our manuscript)
Hoffmann A, Sportelli V, Ziller M, Spengler D. Epigenomics of Major Depressive Disorders and Schizophrenia: Early Life Decides. Int J Mol Sci. 2017;18(8):1711. Published 2017 Aug 4. doi:10.3390/ijms18081711
and about nutritional micronutrients, given as therapeutical adjuvants in patients with depressive disorder (reference 19 from the revised version of our manuscript)
Quan, Z.; Li, H.; Quan, Z.; Qing, H. Appropriate Macronutrients or Mineral Elements Are Beneficial to Improve Depression and Reduce the Risk of Depression. Int. J. Mol. Sci. 2023, 24, 7098. https://doi.org/10.3390/ijms24087098